# Research on Road Scene Understanding of Autonomous Vehicles Based on Multi-Task Learning

**DOI:** 10.3390/s23136238

**Published:** 2023-07-07

**Authors:** Jinghua Guo, Jingyao Wang, Huinian Wang, Baoping Xiao, Zhifei He, Lubin Li

**Affiliations:** 1Department of Mechanical and Electrical Engineering, Xiamen University, Xiamen 361005, China; guojh@xmu.edu.cn (J.G.); 19920220156448@stu.xmu.edu.cn (H.W.); bpxiao@stu.xmu.edu.cn (B.X.); hezhifei@stu.xmu.edu.cn (Z.H.); lilubin1998@163.com (L.L.); 2Department of Automation, Xiamen University, Xiamen 361005, China

**Keywords:** autonomous vehicles, visual perception, multi-task learning, traffic object detection, drivable area detection, lane line detection

## Abstract

Road scene understanding is crucial to the safe driving of autonomous vehicles. Comprehensive road scene understanding requires a visual perception system to deal with a large number of tasks at the same time, which needs a perception model with a small size, fast speed, and high accuracy. As multi-task learning has evident advantages in performance and computational resources, in this paper, a multi-task model YOLO-Object, Drivable Area, and Lane Line Detection (YOLO-ODL) based on hard parameter sharing is proposed to realize joint and efficient detection of traffic objects, drivable areas, and lane lines. In order to balance tasks of YOLO-ODL, a weight balancing strategy is introduced so that the weight parameters of the model can be automatically adjusted during training, and a Mosaic migration optimization scheme is adopted to improve the evaluation indicators of the model. Our YOLO-ODL model performs well on the challenging BDD100K dataset, achieving the state of the art in terms of accuracy and computational efficiency.

## 1. Introduction

The compositions of autonomous driving vehicles can be divided into three modules: environmental perception, decision planning, and vehicle control. Environmental perception is the most fundamental part to realize autonomous driving and is one of the critical technologies in intelligent vehicles [1]. The performance of perception will determine whether the autonomous vehicle can adapt to the complex and changeable traffic environment. The research progress of computer vision shows that visual perception will play a decisive role in the development of autonomous driving [2]. Moreover, vision sensors have the advantages of mature technologies, low prices, and comprehensive detection [3,4].

Effective and rapid detection of traffic objects in various environments will ensure the safe driving of autonomous vehicles, but the detection performance is severely limited by road scenes, lighting, weather, and other factors. The development of big data, computing power, and algorithms has continuously improved the accuracy of deep learning.

Great breakthroughs have been made in the automatic driving industry, making the above detection problems expected to be solved. At present, creating deeper learning networks as deep as possible is the main trend in current research [5], and, while significant progress has been made in that direction, the demands on computational power are becoming increasingly demanding. Runtime is becoming very important when it comes to actually deploying applications.

In view of this, aiming at the problems of redundant calculation, slow speed, and low accuracy in the existing perception models, we propose a multi-task model YOLO-Object, Drivable Area, and Lane Line Detection (YOLO-ODL) model based on hard parameter sharing, which realizes the joint and efficient detection of objects, drivable areas, and lane lines, as shown in Figure 1. The accelerated deployment application of the model is also introduced, so as to provide stable and reliable conditions for decision planning and execution control.

At present, the YOLOv5 object detection model has the best tradeoff between detection accuracy and speed [6]. Based on the YOLOv5s model [7], we built the YOLO-ODL multi-task model. YOLO-ODL has a good balance of detection speed and accuracy, achieving state-of-the-art performance on the BDD100K dataset [8], reaching 94 FPS on an RTX 3090 GPU. The detection speed can reach 477 FPS using TensorRT.

The main contributions of this work are as follows:(I)On the basis of the YOLOv5s model, a multi-task model YOLO-ODL based on hard parameter sharing is built to realize joint and efficient detection of traffic objects, drivable areas, and lane lines.(II)The performance of traffic object detection task is improved by adding shallow high-resolution features and changing the size of the output’s feature map.(III)In order to further improve the performance of YOLO-ODL, the weight balance strategy and Mosaic migration optimization scheme are introduced to improve the evaluation indicators of the multi-task model effectively.

The structure of this paper is organized as follows. Section 2 analyzes related work. Section 3 presents the YOLO-ODL multi-task model. Section 4 presents experimental verification results. Section 5 presents conclusions.

## 2. Related Work

In this section, we review related solutions for the above three tasks of object detection, drivable area detection, and lane line detection, respectively, and then introduce some related multi-task learning work.

### 2.1. Object Detection

Transformer-based object detection methods have had dominant performances in recent years. Zhu et al. [9] applied the transformer to YOLO and achieved better detection results. Anchor-based methods are still the mainstream of object detection at present [10]. Their core idea is to introduce anchor boxes, which can be considered as pre-defined proposals, as a priori for bounding boxes, which can be divided into one-stage object detection methods and two-stage object detection methods. As it is necessary to extract object regions from a set of object proposals and then classify them, the two-stage method is less efficient than the one-stage method [11]. The one-stage method uses the one-stage network to directly output the location and category of objects so that it has evident advantages in training and reasoning time. Liu et al. [12] presented a method of detecting objects in images using a single depth neural network SSD, which could improve the detection speed and achieve the accuracy of two-stage detectors at the same time. Lin et al. [13] designed and trained a simple dense detector called RetinaNet that introduced a new loss function Focal Loss, which effectively solved the problem of imbalanced proportions of positive and negative samples in the training process. The subsequent versions of YOLO [7,14,15] made a series of improvement measures to further improve the object detection accuracy on the premise of ensuring the detection speed.

The actual running time of the object detection algorithm is very important for deploying the application online, so it is necessary to further balance the detection speed and accuracy. In addition, there are a large number of small objects in the autonomous driving environment, but, due to the low-resolution of small objects and the lack of sufficient appearance and structure information, small-object detection is more challenging than ordinary-object detection.

### 2.2. Drivable Area Detection

With the rapid development of deep learning technologies, many effective semantic segmentation methods have been proposed and applied to drivable area detection [16]. Long et al. [17] first proposed FCN, which is an end-to-end semantic segmentation network for opening a precedent for using convolutional neural networks to deal with semantic segmentation problems. Badrinarayanan et al. [18] proposed an encoder–decoder semantic segmentation network named SegNet, which is currently widely used. Zhao et al. [19] established PSPNet, a network that extends pixel-wise features to global pyramid pooling features, thereby combining context information to improve detection performance. Unlike the above-mentioned networks, Tian et al. [20] proposed decoder-adopted, data-dependent upsampling (DUpsampling), which can recover the resolution of feature maps from the low-resolution output of the network. Takikawa et al. [21] proposed a new structure GSCNN, which uses a new structure composed of shape branches and rule branches to focus on boundary information and improves the segmentation ability of small objects.

Based on deep learning, semantic segmentation effectively improves detection accuracy. However, due to the increasing number of deep learning network layers, the model becomes more complex, which leads to the low efficiency of segmentation and is difficult to apply to autonomous driving scenarios.

### 2.3. Lane Line Detection

In order to significantly improve the accuracy and robustness of lane line detection, many lane line detection methods based on deep learning have been proposed. Lane line detection methods can be roughly divided into two categories; one is based on classification, the other is based on semantic segmentation. The classification-based lane detection method reduces the size and computation of the model but suffers from a loss in accuracy and cannot detect scenes with many lane lines well [22]. The method based on semantic segmentation classifies each pixel into lane or background. Pan et al. [23] proposed Spatial CNN (SCNN) that replaced the traditional layer-by-layer convolutions with slice-by-slice convolutions, which enabled message passing between pixels across rows and columns in a layer, thereby improving the segmentation ability for long continuous shape structures such as lane lines, poles, and walls. Hou et al. [24] presented a novel knowledge distillation approach named the Self Attention Distillation (SAD) mechanism, which was incorporated into a neural network to obtain significant improvements without any additional supervision or labels. Zheng et al. [25] presented a novel module named REcurrent Feature-Shift Aggregator (RESA) to collect feature map information so that the network could transfer information more directly and efficiently.

Lane line detection based on semantic segmentation can effectively increase detection accuracy. However, with the increase in detection accuracy, the network becomes more complex, and each pixel needs to be classified, so the detection speed needs to be improved.

### 2.4. Multi-Task Learning

Multi-task networks usually adopt the scheme of hard parameter sharing, which consists of a shared encoder and several feature task decoders. The proposed multi-task network [26] has the advantages of small size, fast speed, high accuracy, and can be used for positioning, making it highly suitable for online deployment of autonomous vehicles. Teichmann et al. [2] proposed an efficient and effective feedforward architecture called MultiNet, which combined classification, detection, and semantic segmentation; the approach shares a common encoder and has three branches on which specific tasks are built with multiple convolutional layers. DLT-Net, proposed by Qian et al. [1], and YOLOP, proposed by Wu et al. [27], simultaneously deal with the problems of traffic objects, drivable areas, and lane line detections in one framework. A multi-task network is helpful to improve model generalization and reduce computing costs. However, multi-task learning may also reduce the accuracy of the model due to the need to balance multiple tasks [28].

Models based on multi-task learning need to learn knowledge from different tasks and are highly dependent on the shared parameters of multi-task models. However, the multi-task architectures proposed by most of the current works lack balancing the relationships among the various tasks. Multi-task models are usually difficult to train to achieve the best effect, and unbalanced learning will reduce the performance of a multi-task model, so it is necessary to adopt meaningful feature representation and appropriate balanced learning styles.

## 3. Proposed Method

Considering the advantages of the multi-task model with hard parameter sharing, such as simple structure, high operating efficiency, and low over-fitting risk, a multi-task model YOLO-ODL based on hard parameter sharing is proposed to achieve joint and efficient detection of objects, drivable areas, and lane lines. The structure of the YOLO-ODL model is shown in Figure 2, which consists of one shared encoder (Shared Encoder) and three decoders (Detection Encoder, Drivable Area Encoder, and Lane Line Encoder) to solve specific tasks.

### 3.1. Shared Encoder

Shared Encoder shared neural network parameters and adopted Backbone and FPN structures in the YOLOv5s object detection model. Backbone extracted common image features from scenes, and FPN integrated image features of different scales (i.e., the information required to detect objects, driving areas, and lane lines). To enhance the feature extraction capability of the Shared Encoder, a shallow high-resolution feature with a size of 160 × 96 was added to the FPN structure. The input image size of the model was 640 × 384 × 3, and the Shared Encoder generated feature maps with three sizes of 160 × 96 × 128, 80 × 48 × 64, and 40 × 24 × 128 from bottom up.

### 3.2. Detection Decoder

Detection Decoder included PANet and Detection Head structures in YOLOv5s object detection model, which were used to decode object detection tasks. PANet further integrated features of different scales, and Detection Head adopted convolutional layer to adjust the number of channels, with 1 × 1 kernel size and stride equal to 1. A lot of small objects, such as traffic lights, traffic signs, pedestrians, and vehicles in the distance, exist in autonomous driving scenarios. Shallow high-resolution features are very important to detect small objects [29], so shallow high-resolution features with sizes of 160 × 96 were added to replace the initial deep low-resolution features with sizes of 20 × 12, and, finally, feature maps with three sizes of 160 × 96 × 18, 80 × 48 × 18, and 40 × 24 × 18 were generated. As each grid was responsible for three anchor boxes, there were a total of 61,440 prediction outputs, and each prediction output included four parameters related to the position of the prediction box, one confidence parameter, and one vehicle category parameter, so the output feature map had 18 channels.

### 3.3. Drivable Area Decoder and Lane Line Decoder

Inspired by the DLT-Net [1] and YOLOP [27] models, we adopted semantic segmentation to realize the detection task of drivable areas and lane lines and used the same decoding structure. The purpose of decoding was to transform the intermediate feature map back to the resolution size of the input image while reducing the number of feature map channels, i.e., transform a low-resolution feature map with a size of 160 × 96 × 128 back to a high-resolution feature map with a size of 640 × 384 × 2, two channels corresponded to the number of classifications, and image features are further extracted to generate denser feature maps, and, finally, the semantic probability output of drivable area and lane line segmentation was generated through the Sigmoid layer.

### 3.4. Loss Function

In the training process of a multi-task model, all tasks start to learn at the same time, so it is necessary to correlate the losses of multiple tasks. We defined the total loss as a weighted sum of losses for object detection, drivable area detection, and lane line detection tasks, so as to make the loss scales of each task closer, where object detection included 3 subtasks concerning bounding box, category, and confidence. The total loss was expressed as
(1)$$L_{total}=\sum_{i\in \left\{\begin{array}{l} box,cls,obj,\\ drivable,lane \end{array}\right\}}w_{i}L_{i}\left(\theta_{i}\right)$$
where $\theta_{i}$ is the neural network parameters, and $w_{i}$ and $L_{i}$ are the loss weight and loss function of specific tasks, respectively.

Remark 1: (1) uses a weight balancing scheme to achieve dynamic adjustment of the weight parameters so as to balance the relationship between individual tasks. The local minima of different tasks in multi-task learning are in different locations, and, by interacting with each other, they can help to escape the local minima, and this paper uses a weight balancing scheme to achieve dynamic adjustment of the weight parameters to minimize the overall loss.

Bounding box prediction is about the judgment of regression parameters, which is a regression problem. From the traditional Smooth L1 to CIoU (Complete IoU) regression loss function, the prediction speed and accuracy of the bounding box have been greatly improved. Therefore, the CIoU loss function is adopted in this paper, and the CIoU loss is calculated by using prediction frames and real frames. The important geometric factors considered by the CIoU Loss function include the overlapping area, center distance, and aspect ratio of the prediction frame and the real frame. The formula for the loss of the bounding box $L_{box}$ is as follows
(2)$$\begin{array}{ll} L_{box}=1-CIoU= & \frac{\rho^{2}\left[\left(\hat{x},\hat{y}\right),\left(x,y\right)\right]}{c^{2}}\\ & +\frac{v^{2}}{1-IoU+v}+1-IoU \end{array}$$
(3)$$v=\frac{4}{\pi^{2}}{\left(\arctan\frac{w}{h}-\arctan\frac{\hat{w}}{\hat{h}}\right)}^{2}$$
where the coordinates of the center point and width and height of the predicted frame are $\left(\hat{x},\hat{y}\right)$ and $\left(\hat{w},\hat{h}\right)$, respectively, and the coordinates of the center point and width and height of the real frame are (x,y) and (w,h), respectively. *ρ* represents the Euclidean distance between the center point of the predicted frame and the real frame; *c* represents the diagonal length of the smallest outer enclosing rectangle of the predicted frame and the real frame; and *v* measures the similarity of the aspect ratio between the predicted frame and the real frame.

Confidence prediction is to judge whether the boundary box contains the target, and category prediction is to judge the target category in the boundary box, both of which belong to classification problems. In this paper, the Binary Cross Entropy (BCE) loss function was used to calculate the classification loss, and the BCE loss function required the input data to be in the range of [0, 1], so the Sigmoid function needed to be used to standardize the input data before input. Assuming that the input image was divided into S × S grids, and each grid was responsible for B prior boxes, the calculation formula of confidence loss $L_{obj}$ and classification loss $L_{cls}$ was as follows
(4)$$\left\{\begin{array}{l} L_{BCE}\left(i,j\right)=-C\left(i,j\right)\cdot \log\hat{C}\left(i,j\right)-\left(1-C\left(i,j\right)\right)\cdot \log\left(1-\hat{C}\left(i,j\right)\right)\\ L_{obj}=\frac{1}{num\left(I_{ij}^{obj}\right)}\sum_{i=0}^{S^{2}}\sum_{j=0}^{B}I_{ij}^{obj}L_{BCE}^{obj}+\frac{1}{num\left(I_{ij}^{noobj}\right)}\sum_{i=0}^{S^{2}}\sum_{j=0}^{B}I_{ij}^{noobj}L_{BCE}^{noobj} \end{array}\right.$$
(5)$$\begin{array}{ll} L_{cls}= & -\frac{1}{nc\cdot num\left(I_{ij}^{obj}\right)}\sum_{i=0}^{S^{2}}I_{ij}^{obj}\sum_{nc\in classes}\left[P\left(i,j,nc\right)\cdot \log\hat{P}\left(i,j,nc\right)\right.\\ & \left.+\left(1-P\left(i,j,nc\right)\right)\cdot \log\left(1-\hat{P}\left(i,j,nc\right)\right)\right] \end{array}$$
where *C* represents the confidence truth label; $\hat{C}$ represents the confidence level of the prediction; *obj* represents the prediction box corresponds to the target object; *noobj* represents the prediction box does not correspond to the target object; $I_{ij}^{obj}$ represents that if the *j*-th prior box of the *i*-th grid is responsible for the target, its value is 1, otherwise it is 0; $num(I_{ij}^{obj})$ represents the number of features corresponding to a target; $I_{ij}^{noobj}$ represents that if the *j*-th prior box of the *i*-th grid is not responsible for the target, its value is 1, otherwise it is 0; $num(I_{ij}^{noobj})$ represents the number of features corresponding to no target; *nc* represents the number of target detection categories number; *P* represents the category true label; $\hat{P}$ represents the predicted category; and the left and right terms in the $L_{obj}$ formula are the confidence losses for positive and negative samples, respectively.

Semantic segmentation is the classification of each pixel in the image, which belongs to the classification problem. Binary cross entropy (BCE) loss function was used to calculate, and Sigmoid function was also used to convert the input data of BCE loss function to [0, 1], where the loss of the whole graph was the average of the loss of each pixel. Semantic segmentation loss was composed of driving area detection loss $L_{drivable}$ and lane line detection loss $L_{lane}$, and the same loss function was used to calculate segmentation loss; the calculation formula is as follows:(6)$$\begin{array}{ll} L_{drivable}=L_{lane}= & -\frac{1}{W\times H}\sum_{m=1}^{W}\sum_{n=1}^{H}\left[S\left(m,n\right)\cdot \log\hat{S}\left(m,n\right)\right.\\ & \left.+\left(1-S\left(m,n\right)\right)\cdot \log\left(1-\hat{S}\left(m,n\right)\right)\right] \end{array}$$
where *W* and *H* represent the width and height of the final output feature map of the segmentation model; *S* represents the semantic segmentation true label; and $\hat{S}$ represents the predicted semantic information.

The bounding box prediction belongs to the regression problem, $L_{box}$ adopts the CIoU loss function, and other tasks belong to the classification problem, using the BCE loss function.

Due to the uncertainty and complexity of different tasks, the multi-task model often appears as a dominant task in the training stage, resulting in the phenomenon of unbalanced training. The performance of the multitask model depends on the weight selection of loss to a large extent, whereas manually adjusting the weight of loss is a time-consuming and laborious process. Cipolla et al. [30] introduced homoscedastic uncertainty to balance multiple tasks and added a learnable noise parameter σ to the loss of each task so that the multi-task network could automatically adjust the weight parameters during training, so as to balance various tasks. Compared with the loss-weighted summation method, this balancing method not only eliminated the need to manually adjust the weight parameters but also improved the joint training performance of the multi-task network. The multi-task loss we finally adopted was shown as
(7)$$\underset{\theta_{i},\sigma_{i}}{\min}\left(\sum_{i\in \left\{\begin{array}{l} box,cls,obj,\\ drivable,lane \end{array}\right\}}\frac{1}{2\sigma_{i}^{2}}L_{i}\left(\theta_{i}\right)+\log\left(1+\sigma_{i}^{2}\right)\right)$$

## 4. Experimental Validation

### 4.1. Experimental Setup

(1)Dataset: The proposed model dataset was trained and tested on the BDD100K [8], a challenging dataset that included a diverse set of driving data under various cities, weather conditions, times, and scene types. The BDD100K dataset also came with a rich set of annotations, including object bounding boxes, drivable areas, and lane markings, with a total of 100 K images with resolution sizes of 1280 × 720, including training (70 K), validation (10 K), and testing (20 K) sets. As the test set was not labeled, we tested the proposed model on the verification set.(2)Metrics: The object detection performance was evaluated by Recall and Mean Average Precision (mAP). mAP50 (mean Average Precision) represented the average precision value of the Intersection over Union (IoU) threshold of 0.5. The drivable area detection performance was evaluated by MIoU [31]. Lane line detection performance was evaluated by pixel accuracy and IoU of lanes [24].(3)Implementation Details: The experimental environment used the Ubuntu 18.04 operating system; a GeForce RTX 3090 graphics card for computing, with 24 GB video memory size; the CPU configuration was an Intel i7-11700K @ 3.60 GHz; the CUDA version was 11.1; the PyTorch version was 1.8.0; and the Python version was 3.8. All experiments were carried out in the same experimental environment with the same training parameters. The initial learning rate was set to 0.001, the weight decay was 0.0005, the momentum was 0.937, and the Adam optimizer was used for optimization training. We adjusted the input size of the model to 640 × 384 to speed up the model and normalize the input image at the same time.

Since we only used BDD100K data set for training, the data were augmented in order to avoid overfitting and improve the generalization ability of the model. The data augmentation adopted included rotating, scaling, translation, color space augmentation, and left–right flipping. The data augmentation process is shown in Figure 3.

### 4.2. Experimental Evaluation

The object detection, drivable area, and lane line branches of the decoder were trained separately, and then the three decoder branches above were trained jointly and compared with various public models. The experiments showed that the single-task model proposed was competitive, and the jointly trained multi-task model reached the state-of-the-art level.

(1)Object Detection Branch: In order to compare with other public models, we filtered out the vehicle class annotations from the BDD100K dataset and trained the single-task model YOLO-Object Detection (YOLO-O) for object detection.

Remark 2: The proposed YOLO-O was improved on the basis of the YOLOv5s model, adding shallow high-resolution features to effectively improve the ability of the target detection, and the detection accuracy was significantly higher than YOLOv5s.

Table 1 shows the comparison of object detection performance. It can be seen that YOLO-O had a higher recall value and was 24.6% and 3% higher than Faster R-CNN and YOLOv5s in terms of mAP50, respectively. It was further proved that the improved object detection model could effectively improve the performance of object detection.

(2)Drivable Area Branch: We uniformly labeled the drivable areas (area/drivable) and the alternative driving areas (area/alternative) in the BDD100K dataset as drivable areas and trained the single-task model YOLO-Drivable Area Detection (YOLO-D) for drivable area detection.

Table 2 shows the comparison of drivable area detection performance. It can be seen that YOLO-D was 23.7% and 2.8% higher than ERFNet and PSPNet in terms of MIoU, respectively.

(3)Lane Line Branch: As the lanes of the BDD100K dataset were labelled by two lines, we used the ENet-SAD [24] method to calculate the lane center line and trained the single-task model YOLO-Lane Line Detection (YOLO-L) for lane line detection.

Remark 3: The proposed YOLO-D and YOLO-L models combined the advantages of the improved YOLOv5s model to expand the semantic segmentation model and used rich data enhancement (see Figure 3 for details) to effectively avoid over fitting and improve the generalization ability of the model, which had advantages in the detection of driving areas and lane lines.

Table 3 shows the comparison of lane line detection performance, where CGAN-L adopted a conditional generative adversarial network to detect lane lines. Although YOLO-L was 1.9% lower than CGAN-L in terms of IoU, it was much higher than the other four lane line detection models in terms of accuracy.

(4)YOLO-ODL: We trained the YOLO-ODL multi-task model to realize the joint detection of objects, drivable areas, and lane lines. Table 4 shows the comparison of YOLO-ODL performance indicators. It can be seen that the runtime of the YOLO-ODL model had evident advantages over the total runtimes of the three single-task models, but it was a little worse in terms of evaluation indicators. Therefore, we adopted the weight balancing scheme of Formula (2) to train the YOLO-ODL model so that the model could automatically balance the multi-task weights. Mosaic data augmentation effectively improved the accuracy of object detection [15], so the migration optimization scheme shown in Figure 4 was adopted. Mosaic data augmentation was added, and then the knowledge transfer learned from the YOLO-O object detection model was applied to the multi-task model.

As shown in Table 4, YOLO-ODL (+ weight balance) improved the performances of the three tasks, which could better balance each task. YOLO-ODL (+ weight balance and migration optimization) further improved the evaluation indicators of the model, especially in object detection, thus proving the effectiveness of weight balance and migration optimization. In general, the optimized YOLO-ODL had advantages over the three single-task models.

In view of the variability of weather and the complexities of road scenes, the YOLO-ODL model was tested under multi-weather and multi-scenario road conditions, respectively, to further embody the robustness of the acceleration model. As shown in Figure 5 and Figure 6, YOLO-ODL had strong robustness in multi-weather and multi-scenario road conditions and could well detect traffic objects, drivable areas, and lane lines.

The comparison of the multi-task model detection performance is shown in Table 5. As the experimental test platforms were different, the detection speeds of MultiNet and DLT-Net were much lower than that of YOLOP, and the detection speeds of MobileNetV3 and RegNetY were comparable to YOLOP, this paper only tested the FPS of YOLOP and YOLO-ODL. The experimental results showed that the detection effect and the detection speed of YOLO-ODL were superior to other multi-task models.

Remark 4: The proposed YOLO-ODL model combined the advantages of the improved YOLOv5s model and further improved the accuracy of the model by adopting a weight balance and migration optimization scheme. Therefore, in the aspect of multi-task detection, the proposed YOLO-ODL was superior to other multi-task models.

As shown in Figure 7, Figure 8 and Figure 9, we compared the detection effects of YOLO-ODL and YOLOP on three tasks. Object detection results are shown in Figure 7. YOLOP was prone to false detections and missed detections, mistakenly detecting complex house backgrounds as vehicles, missing the detection of vehicles in the distance. Figure 8 shows drivable area detection results. YOLOP had incomplete detection and many false detection areas, whereas the drivable area detected by YOLO-ODL was more accurate and complete. Lane line detection results are shown in Figure 9. The lane lines detected by YOLO-ODL were more accurate and continuous, but the lane lines detected by YOLOP were discontinuous, and there were more noise areas.

TensorRT was used to accelerate the multi-task model and further improve the deployment performance of the model. As shown in Table 6, YOLO-ODL, after the accelerated deployment of TensorRT-FP16, was not only smaller in size but also greatly improved the inference speed. In addition, we also provided Python and C++ APIs to run the model. YOLO-ODL’s acceleration ratio after TensorRT-FP16 C++ accelerated deployment reached 5.07, and the detection speed reached 477 FPS, which further verified the efficiency of TensorRT.

## 5. Conclusions

In view of the limited computing resources of autonomous vehicles, a multi-task model based on hard parameter sharing was built, which consisted of a shared encoder and three task-specific decoders, which greatly improved computing efficiency. In order to balance the various tasks of the multi-task model, we introduced the loss balance strategy to the multi-task model so that the multi-task model could automatically adjust the weight parameters during training, and we adopted the Mosaic transfer optimization scheme to improve the evaluation index of the multi-task model. Then, the multi-task model was trained and tested to prove the ability of the model to detect the targets, drivable areas, and lane lines and verify the effectiveness of the loss balance strategy and the Mosaic migration optimization scheme. Finally, compared with the latest multi-task network model, it was proved that the proposed YOLO-ODL multi-task model could achieve state-of-the-art performance.

## Figures and Tables

**Figure 1 sensors-23-06238-f001:**
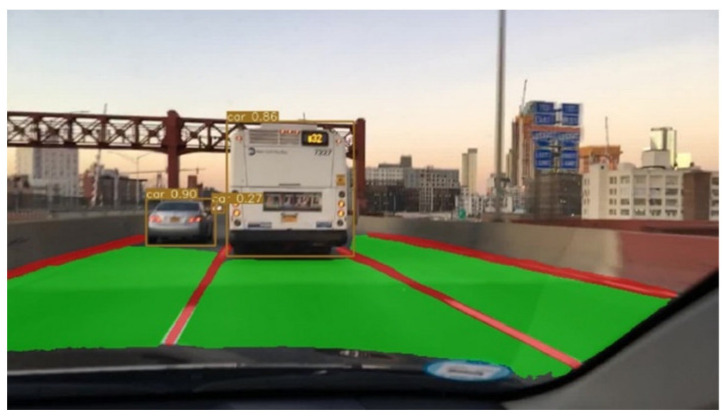
Our road detection.

**Figure 2 sensors-23-06238-f002:**
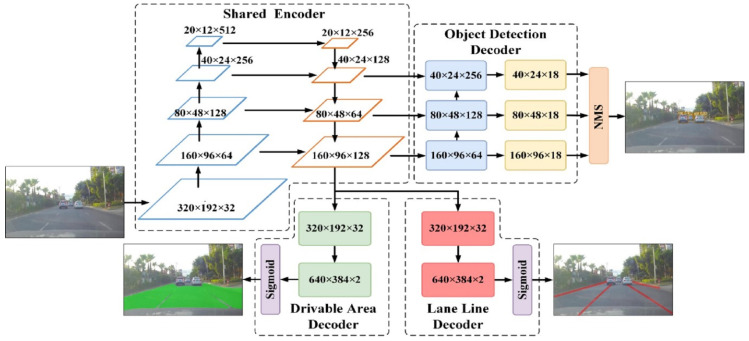
Structure of the proposed YOLO-ODL model.

**Figure 3 sensors-23-06238-f003:**
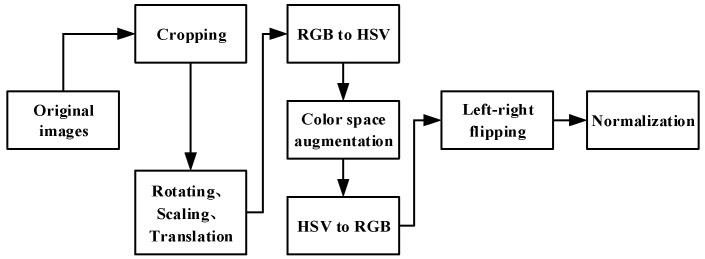
Data augmentation.

**Figure 4 sensors-23-06238-f004:**
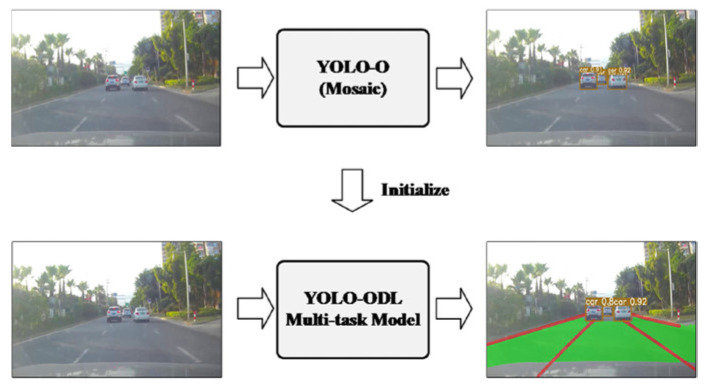
Migration optimization.

**Figure 5 sensors-23-06238-f005:**
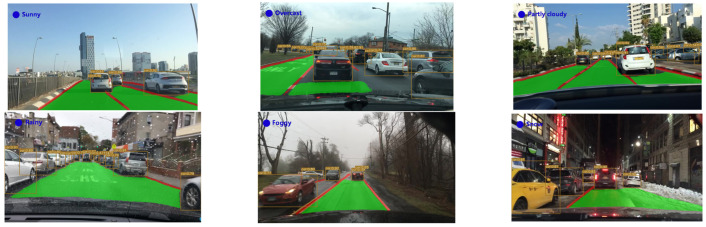
Detection results of YOLO-ODL in multi-weather road conditions.

**Figure 6 sensors-23-06238-f006:**
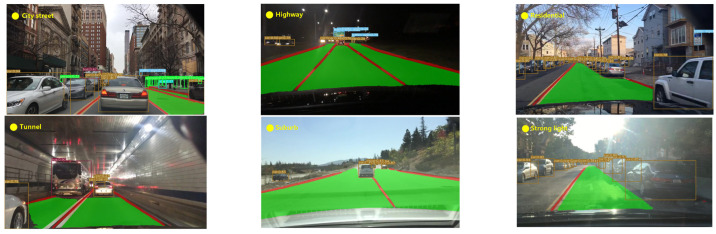
Detection results of YOLO-ODL in multi-scenario road conditions.

**Figure 7 sensors-23-06238-f007:**
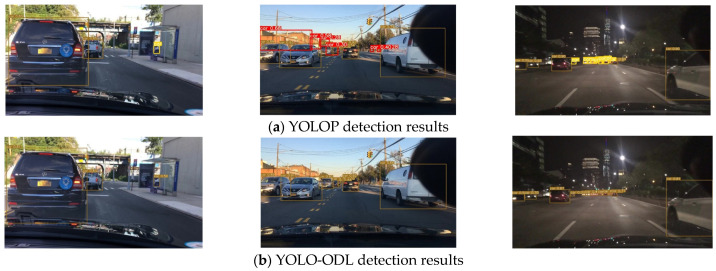
Comparison of YOLO-ODL and YOLOP object detection results. The red box in the figure is the false detection box, and the yellow box is the missed detection box.

**Figure 8 sensors-23-06238-f008:**
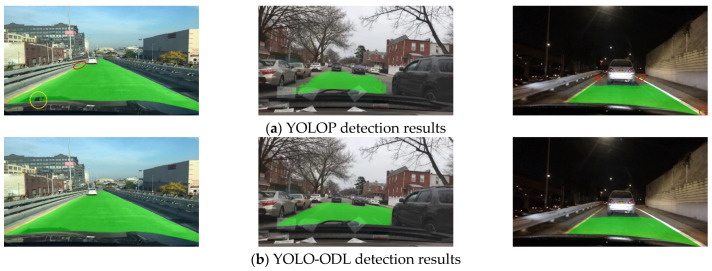
Comparison of YOLO-ODL and YOLOP drivable area detection results. The red circle in the figure is the false detection area, and the yellow circle is the missed detection area.

**Figure 9 sensors-23-06238-f009:**
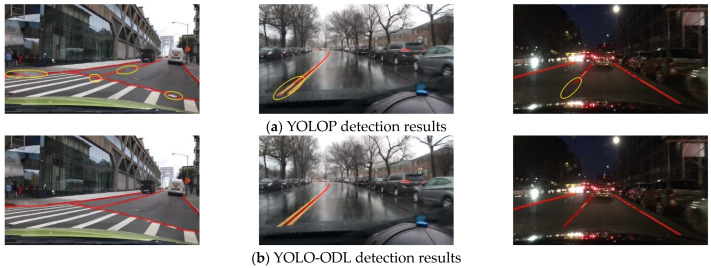
Comparison of YOLO-ODL and YOLOP lane line detection results. The yellow circle is the missed detection area.

**Table 1 sensors-23-06238-t001:** Object Detection Performance Comparison.

Model	Recall	mAP50
Faster R-CNN [1]	77.2%	55.6%
YOLOv5s [27]	86.8%	77.2%
YOLO-O	**94.4%**	**80.2%**

**Table 2 sensors-23-06238-t002:** Drivable Area Detection Performance Comparison.

Model	MIoU
ERFNet [1]	68.7%
PSPNet [27]	89.6%
YOLO-D	**92.4%**

**Table 3 sensors-23-06238-t003:** Lane Line Detection Performance Comparison.

Model	Accuracy	IoU
SCNN [27]	35.8%	15.8%
ENet-SAD [24]	36.6%	16.0%
CGAN-L [32]	57.2%	**30.0%**
SALMNet [33]	58.3%	25.1%
YOLO-L	**80.0%**	28.1%

**Table 4 sensors-23-06238-t004:** YOLO-ODL Performance Indicators Comparison.

Model	Recall	mAP50	MIoU	Accuracy	IoU	Runtime
YOLO-O	94.4%	80.2%	-	-	-	7.3 ms
YOLO-D	-	-	**92.4%**	-	-	6.9 ms
YOLO-L	-	-	-	**80.0%**	**28.1%**	6.9 ms
YOLO-ODL	94.2%	79.7%	92.3%	75.0%	26.8%	10.6 ms
+ weight balance	94.3%	80.0%	92.4%	75.2%	26.9%	-
+ weight balance and migration optimization	94.5%	81.0%	92.4%	75.4%	27.5%	-

**Table 5 sensors-23-06238-t005:** Multi-task Model Performance Comparison.

Model	Recall	mAP50	MIoU	Accuracy	IoU	FPS
MultiNet [1]	81.3%	60.2%	71.6%	-	-	-
DLT-Net [1]	89.4%	68.4%	72.1%	-	-	-
YOLOP [27]	89.2%	76.5%	91.5%	70.5%	26.2%	75
MobileNetV3 [16]	82.7%	75.2%	89.8%	75.9%	28.8%	-
RegNetY [16]	86.1%	77.5%	91.9%	**76.9%**	**33.8%**	-
YOLO-ODL	**94.2%**	**79.7%**	**92.3%**	75.0%	27.5%	**94**

**Table 6 sensors-23-06238-t006:** Accelerated Deployment of YOLO-ODL.

Framework	Programming Language	Model Size	FPS	Acceleration Ratio
PyTorch	Python	22.2 MB	94	-
TensorRT-FP16	Python	15.3 MB	341	3.63
TensorRT-FP16	C++	15.3 MB	477	5.07

## Data Availability

The data used to support the finding of this study are available from the corresponding author upon request.

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
