# Peer review of "Research on Road Scene Understanding of Autonomous Vehicles Based on Multi-Task Learning"

_sensors, 2023, doi:10.3390/s23136238_

Round 1

Reviewer 1 Report

The paper proposed a multi-task detection method for autonomous vehicle based on YOLO V5. It is an interesting work. However, I have some concerns needed to be addressed.

(1) The listed references are very old and cannot illustrate the state-of-the-art work. The authors should update the literature review to most recent work (2023).

(2) The titles of 3.2 and 3.3 should be "Detection Decoder" and "Drivable Area Decoder and Lane Line Decoder", respectively.

(3) Usually, the math symbols with hat are prediction results. The authors should check all the math symbols in the manuscript carefully.

(4) In line 202, the words or symbols in "The ... is expressed as" is missed.

(5) I think the section 3.5 is not necessary to presented in the manuscript.

(6)  More recent work on multi-task detection should be compared with the proposed method.

(7) The experiments on more datasets should be conducted.

NA

Author Response

Reviewer: 1

Question 1: The listed references are very old and cannot illustrate the state-of-the-art work. The authors should update the literature review to most recent work (2023).

Reply: Thanks for the reviewer’s comments. References for 2023 have been added to the text.

Question 2: The titles of 3.2 and 3.3 should be "Detection Decoder" and "Drivable Area Decoder and Lane Line Decoder", respectively.

Reply: Thanks for the reviewer’s comments. It has been modified in the text.

Question 3: Usually, the math symbols with hat are prediction results. The authors should check all the math symbols in the manuscript carefully.

Reply: Thanks for the reviewer’s comments. The math symbols with hat in equations (4), (5), and (6) have been expressed as predictions, and other math symbols have been checked.

Question 4: In line 202, the words or symbols in "The ... is expressed as" is missed.

Reply: Thanks for the reviewer’s comments. The missing total loss has been added in the article.

Question 5: I think the section 3.5 is not necessary to presented in the manuscript.

Reply: Thanks for the reviewer’s comments. Section 3.5 has been removed from the text.

Question 6: More recent work on multi-task detection should be compared with the proposed method.

Reply: Thanks for the reviewer’s comments. The proposed method has been compared in the paper with recent work on multitask detection, with the work on multitask detection published in March 2023 in the red box.

Question 7: The experiments on more datasets should be conducted.

Reply: Thanks for the reviewer’s comments. The experimental data used in this paper are from the BDD100K dataset, a more challenging dataset made public by the University of California, Berkeley. The BDD100K dataset contains 100,000 high-definition video sequences, each 40 seconds long and 30 FPS, so there are 120 million raw data volumes, which is much larger than other data sets, and the data set also covers different cities, driving scenarios, weather conditions and time periods. Therefore, the BDD100K data set adopted in this paper can test the model well.

Reviewer 2 Report

“This manuscript uses a well-known image recognition method “YOLO” to detect traffic object, drivable area and lane line. They tried to adjust the parameters to achieve better accuracy using a benchmark dataset. In my opinion, the overall contribution, structure and quality of the manuscript are not strong enough to make it suitable for publication in such reputable journal. Also, there are not enough justification to the research motive or its idea importance to the topic research community.”

English very difficult to understand/incomprehensible

Author Response

Question 1: There are not enough justification to the research motive or its idea importance to the topic research community.”

Reply: Thanks for the reviewer’s comments. Aiming at the problems of redundant calculation, slow speed and low accuracy in the existing perception models, we propose a multi-task model YOLO-Object, Drivable Area and Lane Line Detection (YOLO-ODL) based on hard parameter sharing, which realizes the joint and efficient detection of object, drivable area and lane line. The accelerated deployment application of the model is also introduced, so as to provide stable and reliable conditions for decision planning and execution control.

The main contributions of this work are as follows:

(â… ) On the basis of YOLOv5s model, a multi-task model YOLO-ODL based on hard parameter sharing is built to realize joint and efficient detection of traffic object, drivable area and lane line.

(â…¡) The performance of traffic object detection task is improved by adding shallow high-resolution features and changing the size of the output feature map.

(III) In order to further improve the performance of YOLO-ODL, the weight balance strategy and Mosaic migration optimization scheme are introduced to improve the evaluation indicators of the multi-task model effectively.

Reviewer 3 Report

Dear authors,

in my opinion, the paper is well written, the problem has been correctly described, it contains enough references and interesting results have been presented.

But, I would like to include some comments to improve some parts of it:

- In section 3, Proposed Method, here more details should be included. E.g formulas (1) to (7), that model the system, should be justified. In this work, these formulas are included but it is neceesary to know why this concepts have been modeled in this way.

- In section 4, Experimental Validation, to be honest, the system presents a very good results compared with other algorithms but the methodoly and how the experiments have been set up it is not clear. I recommend to include more details about this experiments.

- In  section5, Conclusions, the conclusions are very poor and not very explanatory. I suggest that a more general overview of the contributions of this work be presented.

Best regards.

Dear authors,

from my point of view, the English used in the manuscript is correct and the concepts and ideas can be clearly understood.

Regards.

Author Response

Question 1: In section 3, Proposed Method, here more details should be included. E.g formulas (1) to (7), that model the system, should be justified. In this work, these formulas are included but it is neceesary to know why this concepts have been modeled in this way.

Reply: Thanks for the reviewer’s comments. Formulas (1) to (7) are the total loss function and each subtask loss function. The selection of loss function has been further explained in the paper.

Question 2: In section 4, Experimental Validation, to be honest, the system presents a very good results compared with other algorithms but the methodoly and how the experiments have been set up it is not clear. I recommend to include more details about this experiments.

Reply: Thanks for the reviewer’s comments. The method proposed in this paper is based on the YOLOv5s model, and we have explained the experimental method in detail in this paper

Question 3: In section5, Conclusions, the conclusions are very poor and not very explanatory. I suggest that a more general overview of the contributions of this work be presented.

Reply: Thanks for the reviewer’s comments. The conclusion part has been rediscussed in the paper.

Round 2

Reviewer 1 Report

Please check all the symbols in the text and equations, e.g., 

(1) In Equations (4) and (5), line 240, where are the C and C hat? 

(2) In Equation (6) and line 259-260, where are the S and S hat?

(3) The ground truth and prediction should have similar forms, in Equations (2) and (3), the ground truth is represented as X^gt and the predicted value is represented as X^pred, but the Equations (4)-(6), the  the ground truth is represented as X and the predicted value is represented as X^, they should have similar forms.

Author Response

The manuscript is revised based on the comments. 

Reviewer 2 Report

I noticed a considerable improvement in the manuscript. therefore I will change my recommendation to accept with major revisions. However, the authors need to make some extra effort in the experimental section, especially with the presentation with plots to show the performance of their methodology.

Also, they need to add some review about the sensor location problem I suggest these two articles to refer to:S "On-road vehicle detection: A review." IEEE transactions on pattern analysis and machine intelligence 28.5 (2006): 694-711. and  "Traffic sensor location problem: Three decades of research." Expert Systems with Applications (2022): 118134.

.

An additional language revision is needed to raise the manuscript's quality.

Author Response

The manuscript is revised based on the comments of reviewers.

Round 3

Reviewer 2 Report

None.